# OpenReview forum: "ExpertLongBench: Benchmarking Language Models on Expert-Level Long-Form Generation Tasks with Structured Checklists"
_ICLR.cc/2026/Conference — ICLR 2026 Poster_

### Official Review · Reviewer_LZpR · 2025-10-27

**Soundness:** 3
**Presentation:** 3
**Contribution:** 3
**Rating:** 6
**Confidence:** 4

**Summary:**

This paper introduces `ExpertLongBench`, a long-form benchmark curated by experts across 11 tasks and 9 domains. The authors provide a detailed walkthrough of the benchmark creation process and propose an evaluation framework called `CLEAR`, and evaluate 13 LLMs on `ExpertLongBench` using the `CLEAR` framework.

**Strengths:**

- The paper is well articulated, and I appreciate the long-context benchmark provided in the paper. I strongly resonate with the authors' motivation, as long-context evaluation techniques are indeed understudied. Additionally, the benchmark covers some practical tasks, whose data is not readily available.
- I also acknowledge the capital and manual effort invested in this benchmark creation process, and I believe this will be a solid addition to the standard evaluation suite for LLMs.
- I also appreciate the authors' choice to keep the test set private to avoid contamination.

**Weaknesses:**

- I don't find any glaring technical errors in the paper. Not a weakness per se, by most of the content of this paper is about the benchmark creation process using experts-in-the-loop, while the second half discusses a standard benchmarking process on this benchmark following the `CLEAR` framework. The technical takeaways are minimal compared to the data contribution.
- Again, not a weakness per se, but a massive chunk of the important information is stowed away in the appendix, and it is a hassle to refer back and forth.
- I do have some comments, suggestions, and queries for the authors below, which I request the authors to clarify.
- The benchmark might not be extendable to other domains or languages, given the expert-in-the-loop methodology.

**Questions:**

- Since the benchmark is so domain-specific, why not conduct some sort of RAG-based evaluations? I do understand that this makes the evaluation a two-phase process (retrieval evaluation and then generation evaluation, given the retrieval). However, I feel it might be unfair to expect the LLM to know intricate and nuanced domain-specific information, which is arcane to common users, and it might reflect the true abilities of the model.
- I suggest that some reasoning models also be evaluated on this benchmark, especially given Figure 5. It would be interesting to see if test-time scaling improves the performance on harder problems. I feel this will be an interesting takeaway, which might show the efficacy of reasoning on non-math/code and factual, knowledge-intensive tasks.
- It would also be interesting to see how the rankings of these LLMs on `ExpertLongBench` compare to `MMLU`/`GPQA` and `LMArena` rankings. The latter is generally treated as the empirical ranking, while MMLU is _(unfortunately)_ reported as a standard benchmark, and many model builders make their choice solely based on it. Hence, a difference in ranking would further motivate the need for domain-specific and long-context evaluations. However, I do acknowledge that most of the questions in MMLU and LMArena might be quite small compared to `ExpertLongBench`, which specializes in long-context queries. This won't be an apples-to-apples comparison.
- If possible, I urge the authors to add a small study on LLM-evaluator's self-bias, as this benchmark provides an ideal ground for it. In an $N \times N$ matrix, we can have the performance of LLMs as evaluated by every LLM, including itself. On this hard benchmark, if we notice the diagonal consistently higher, it would be a clear example of self-bias, where the model scores itself higher despite a lack of knowledge of the domain/incorrect answers. I hope this is clear.

---

> ### Author Response · Authors · 2025-11-22
> **Response to Reviewer LZpR (1/4)**
>
> Dear Reviewer LZpR,
>
> We genuinely thank you for the time and effort you devoted to reviewing our paper. Your detailed and insightful suggestions are of great value to us as we work to improve the quality of our paper. In the following section, we address the concerns you raised:
>
> > **Weakness**: The benchmark might not be extendable to other domains or languages, given the expert-in-the-loop methodology.
>
> **Response**: We acknowledge that there is an inherent trade-off between scalability and reliability. In designing our benchmark using expert-in-the-loop methodology, we prioritized reliability, as we believe it is critical to ensure that evaluation results meaningfully reflect the model’s true capabilities. If the evaluation is not reliable, it becomes difficult to interpret the results or draw meaningful conclusions about model performance.
>
> However, we understand the reviewer’s concern regarding extendability. It is indeed possible to reduce the expert load for a balance between reliability and expandability:
> - **Protocol-refinement strategy** (mentioned in page 4 “Expert-guided Rubric Design” paragraph). Many practical domain-specific tasks are governed by well-established professional standards. For such tasks, rubric creation primarily involves adapting existing protocols rather than designing criteria from scratch, substantially reducing expert workload.
> - **Human–LLM collaborative rubric design**. LLMs can assist experts by generating initial checklists or draft rubrics based on task descriptions and high-quality human reference outputs. Experts would then validate and refine these drafts, reducing the required expert hours while preserving domain rigor. Existing rubrics in our benchmark would further serve as in-context examples, enabling LLMs to propose increasingly accurate initial drafts over time.
> - **Active collaboration with domain communities**. We plan to expand ExpertLongBench through multi-disciplinary, open collaboration, inviting domain researchers and practitioners to contribute tasks, provide feedback, and help refine rubrics. This shared stewardship further distributes the effort and enhances the benchmark’s coverage and credibility.
>
> We agree that expert benchmarking is inherently difficult and this is one of the reasons why our work is valuable. ExpertLongBench addresses a gap that simpler QA or multiple-choice datasets cannot: evaluating end-to-end, expert-level long-form generation. By combining careful task selection, protocol-based rubric design, human–LLM collaboration, and a dynamic refresh mechanism, we believe this benchmark can scale sustainably without requiring hundreds of expert hours for each new task.
>
> **For expanding to other languages**, Our benchmarking method and framework are language-agnostic, and all code, rubrics, and pipelines are released to support extension to other languages. We will establish a community contribution mechanism on the benchmark website, inviting researchers from the community to submit high-quality datasets that align with the benchmark's objectives, including data in other languages.
>
> We appreciate the reviewer’s perspective and will incorporate these clarifications into the revised version.

---

> ### Author Response · Authors · 2025-11-22
> **Response to Reviewer LZpR (2/4)**
>
> > **Question 1**: Since the benchmark is so domain-specific, why not conduct some sort of RAG-based evaluations? I do understand that this makes the evaluation a two-phase process (retrieval evaluation and then generation evaluation, given the retrieval). However, I feel it might be unfair to expect the LLM to know intricate and nuanced domain-specific information, which is arcane to common users, and it might reflect the true abilities of the model.
>
> **Response 1**: Our primary goal with this benchmark is to assess whether current frontier models possess the expert knowledge and practical skills needed to solve complex, real-world expert tasks. Nonetheless, we also ran an additional experiment on the most challenging task (T2) where the models were given fine-grained, explicit guidance about the task requirements as specified in Table 61. In this setup, we specified the exact elements that must appear in the model’s response and clarified what each criterion entails.
>
> Although this scaffolding improved performance, the gains remained far below acceptable standards. Without targeted prompting, GPT-4o and Gemini-2.0-Flash achieved F1-scores of 6.2 and 7.9, respectively. With explicit prompting, their scores increased to 32.5 and 28.6, yet still substantially short of the proficiency required for practical deployment. This highlights the depth and difficulty of our benchmark and the significant gap that remains even when models are provided with detailed task specifications and nuanced information.
>
> As for a RAG-based evaluation, we implemented a RAG-based agent using GPT-5 with langchain that sequentially retrieves and integrates relevant information from the provided document context to address each task. We implemented retrieval by embedding all document paragraphs using OpenAI’s text-embedding-3-large model. At each iteration, GPT-5 generates a query, which is then used to retrieve the most relevant paragraphs; these retrieved passages are subsequently fed back into GPT-5 to support reasoning and task completion.  Because Tasks T1 and T2 involve particularly long documents, where retrieval-augmented methods are especially applicable, we applied this RAG approach to these tasks. The evaluation results (F1-scores) are presented below. We find that the RAG agent underperforms, suggesting that full access to the global document context is crucial for these tasks. Although this generic RAG implementation did not yield performance gains, we believe that a domain- and task-specific retrieval and reasoning strategy could lead to stronger results. However, developing such specialized systems is beyond the scope of our current work.
>
>
> | Task | GPT-5  | GPT-5 RAG Agent |
> |:----:|:-----:|:----------------:|
> | T1   | 27.15 | 24.27           |
> | T2   | 10.34 | 4.85           |
>
> > **Question 2**: I suggest that some reasoning models also be evaluated on this benchmark, especially given Figure 5. It would be interesting to see if test-time scaling improves the performance on harder problems. I feel this will be an interesting takeaway, which might show the efficacy of reasoning on non-math/code and factual, knowledge-intensive tasks.
>
> **Response 2**: We thank the reviewer for this excellent suggestion. Following your recommendation, we evaluated several reasoning models to examine whether test-time scaling improves performance on our expert-level, knowledge-intensive tasks.
> These are the models evaluated:
> - O3-2025-04-16, selected based on the LMArena leaderboard as a top reasoning model.
> - Qwen3-32B, a strong open-source reasoning model representative of current open reasoning-focused systems.
> - Gemini-2.5-Pro, previously included in our paper, for which thinking mode is enabled.
>
> The F1 scores scores across all tasks are shown below:
> | Model            | T1   | T2  | T3  | T4   | T5   | T6   | T7   | T8   | T9   | T10  | T11  | Avg  |
> |------------------|------|-----|-----|------|------|------|------|------|------|------|------|------|
> | Gemini-2.5-Pro   | 25.4 | 10.0|16.9 | 50.2 | 47.9 | 44.0 | 50.4 | 14.9 | 43.2 | 21.1 | 43.3 | 33.4 |
> | o3               | 25.3 | 8.1 | 7.7 | 49.1 | 43.5 | 52.5 | 47.3 | 13.0 | 30.7 | 10.2 | 34.4 | 29.3 |
> | Qwen3-32B        | 17.7 | 3.6 |14.7 | 52.0 | 33.0 | 47.6 | 45.1 | 12.6 | 31.3 | 18.7 | 33.1 | 28.1 |
>
> Results show test-time scaling does not substantially improve domain-specific reasoning and reasoning models do not close the gap to domain experts. In tasks with well-defined professional standards, the performance gap remains large. This underscores the challenge of moving beyond math/code reasoning toward domain-grounded, factual, expert-level long-form generation. We appreciate the reviewer’s suggestion. We will add these results and analysis to the revised version.

---

> > ### Author Response · Authors · 2025-11-22
> > **Response to Reviewer LZpR (3/4)**
> >
> > > **Question 3**: It would also be interesting to see how the rankings of these LLMs on ExpertLongBench compare to MMLU/GPQA and LMArena rankings. The latter is generally treated as the empirical ranking, while MMLU is (unfortunately) reported as a standard benchmark, and many model builders make their choice solely based on it. Hence, a difference in ranking would further motivate the need for domain-specific and long-context evaluations. However, I do acknowledge that most of the questions in MMLU and LMArena might be quite small compared to ExpertLongBench, which specializes in long-context queries. This won't be an apples-to-apples comparison.
> >
> > **Response 3**: We thank the reviewer for this valuable suggestion. Following your recommendation, we compared the model rankings on ExpertLongBench with their standings on MMLU, GPQA, and LMArena (text leaderboard). The results are summarized in the table below. We observe divergence between model performance on ExpertLongBench and their rankings on standard benchmarks. These discrepancies reinforce the reviewer’s point: performance on short-form factual QA (MMLU) or pairwise preference evaluations (LMArena) does not predict performance on complex, domain-specific long-form tasks. This divergence strongly supports the need for domain-grounded, long-form, and professional-task-oriented evaluation, as provided by ExpertLongBench. While, as the reviewer noted, this comparison is not apples-to-apples due to the long-context and long-form nature of our tasks, the ranking differences themselves highlight the shortcomings of relying solely on existing standard benchmarks. We will include this comparison and discussion in the revised version, as it further motivates the importance and necessity of our proposed benchmark.
> >
> > | Model                        | ExpertLongBench Ranking | LMArena Ranking  | MMLU Ranking  | GPQA Ranking  |
> > |------------------------------|:-----------------:|:---------:|:------:|:------:|
> > | Gemini-2.5-Pro               | 1               | 1       | 2    | 3    |
> > | GPT-5                        | 2               | 3       | 1    | 1    |
> > | o3                           | 3               | 2       | 3    | 2    |
> > | Qwen3-32B                    | 4               | 6       | -    | -    |
> > | Gemini-2.0-Flash             | 5               | 5       | 6    | 5    |
> > | GPT-4o                       | 6               | 7       | 5    | 6    |
> > | GPT-4o-mini                  | 7               | 10      | 10   | 9    |
> > | Llama-3.3-70B-Instruct       | 8               | 9       | 8    | 7    |
> > | Mistral-Large-Instruct       | 9               | 11      | 7    | 8    |
> > | Qwen2.5-72B                  | 10              | 12      | -    | -    |
> > | Mistral-Nemo-Instruct        | 11              | -       | 13   | -    |
> > | Llama3.1-8B-Instruct         | 12              | 13      | 12   | -    |
> > | Claude-3.7-Sonnet            | 13              | 4       | 4    | 4    |
> > | Qwen2.5-7B                   | 14              | -       | 11   | -    |
> > | Claude-3.5-Haiku             | 15              | 8       | 9    | 10   |

---

> > > ### Author Response · Authors · 2025-11-22
> > > **Response to Reviewer LZpR (4/4)**
> > >
> > > > **Question 4**: If possible, I urge the authors to add a small study on LLM-evaluator's self-bias, as this benchmark provides an ideal ground for it. In an  matrix, we can have the performance of LLMs as evaluated by every LLM, including itself. On this hard benchmark, if we notice the diagonal consistently higher, it would be a clear example of self-bias, where the model scores itself higher despite a lack of knowledge of the domain/incorrect answers. I hope this is clear.
> > >
> > > **Response 4**: We thank the reviewer for this helpful suggestion and fully agree on the importance of assessing potential self-bias in LLM evaluators. To conduct this analysis, we are evaluating 8 models (Gemini-2.0-Flash, Gemini-2.5-pro, GPT-4o, Mistral-Large-Instruct, Llama-3.1-8B-Instruct, Qwen2.5-72B, Qwen2.5-7B, and Claude-3.5-Haiku) across five tasks (T1, T3, T4, T6, T8), using four evaluator models (Gemini-2.0-Flash, GPT-4o, and Qwen2.5-72B). These models span a wide performance spectrum, and the selected tasks cover diverse domains with varied input and output lengths.
> > >
> > > For each task, we compute performance using all 3 evaluators, derive evaluator-specific rankings, and then measure the Spearman rank correlation between GPT-4o’s rankings and those produced by the other evaluators to assess consistency. The table below reports these correlations for each task. The consistently high correlations indicate strong agreement across evaluators. This is expected because the evaluation focuses on measuring semantic overlap between the model-produced checklists and the ground-truth checklists which is a substantially easier task than solving the underlying expert tasks themselves. As a result, any sufficiently capable evaluator should yield highly consistent rankings.
> > >
> > > | Task | GPT-4o vs Gemini-2.0-Flash | GPT-4o vs Qwen2.5-72B | Gemini-2.0-Flash vs Qwen2.5-72B |
> > > |------|:-----------------------------:|:-------------------------:|:----------------------------------:|
> > > | T1   | 0.6429                      | 0.9762                  | 0.5952                           |
> > > | T3   | 0.9762                      | 0.8333                  | 0.7381                           |
> > > | T4   | 0.9524                      | 0.9762                  | 0.9762                           |
> > > | T6   | 0.8571                      | 1.0000                  | 0.8571                           |
> > > | T8   | 0.9524                      | 0.9286                  | 0.9524                           |
> > >
> > > For the self-bias analysis, we computed the task-wise average performance (over the aforementioned tasks) of GPT-4o, Gemini-2.0-Flash, and Qwen2.5-72B as evaluated by one another, and summarized the results in a table where rows correspond to the evaluator and columns correspond to the model under evaluation. The close alignment of scores across evaluators shows that our evaluation setup exhibits minimal self-bias.
> > >
> > > |                | GPT-4o | Gemini-2.0-Flash | Qwen2.5-72B |
> > > |----------------|:--------:|:-------------------:|:-------------:|
> > > | **GPT-4o**            | 0.2416 | 0.2235            | 0.2225      |
> > > | **Gemini-2.0-Flash**  | 0.2336 | 0.2156            | 0.2139      |
> > > | **Qwen2.5-72B**       | 0.2341 | 0.2221            | 0.2162      |
> > >
> > >
> > > We hope this addresses your concern, and we would greatly appreciate it if you could consider updating your review score. Once again, thank you very much for your insightful and valuable suggestions!

---

> > > > ### Comment · Reviewer_LZpR · 2025-11-24
> > > > **Final Official Comment to the Authors**
> > > >
> > > > Dear Authors,
> > > >
> > > > I sincerely appreciate and acknowledge the additional experiments and detailed findings shared here. I hope these new findings make their way into the paper as well. I am content with the new findings, and I have improved my score $8.0$ at the moment. I feel this is a just evaluation from my end. Thank you!

---

> > > > > ### Author Response · Authors · 2025-11-24
> > > > > **Response to Reviewer LZpR’s Final Official Comment**
> > > > >
> > > > > Dear Reviewer LZpR,
> > > > >
> > > > > We sincerely appreciate your positive feedback and are glad that the additional experiments and analyses addressed your concerns. We will incorporate the new findings into the revised version of the paper. We are also very grateful for your insightful and valuable suggestions, which have been very helpful in improving the overall quality of our work. Thank you for taking the time to review our paper and for improving your score. We truly appreciate your thoughtful and fair evaluation.

---

### Official Review · Reviewer_15Gk · 2025-10-31

**Soundness:** 3
**Presentation:** 3
**Contribution:** 2
**Rating:** 6
**Confidence:** 3

**Summary:**

This paper introduces EXPERTLONGBENCH, a new benchmark designed to evaluate LLMs on expert-level, long-form generation tasks. The benchmark consists of 11 tasks from 9 professional domains that mimic real-world workflows, featuring long contexts (up to 200K tokens) and long-form outputs (up to 5K tokens). The primary contribution is the CLEAR evaluation framework. Instead of direct text-to-text comparison, CLEAR uses expert-designed "rubrics" to create a "checklist" of required information. It then uses an LLM to extract this checklist of information from both the model's output and a human-written reference, and compares these two checklists item-by-item. The paper benchmarks 13 SOTA LLMs, finding that even the best model (Gemini-2.5-Pro) achieves a very low F1 score (~33.4%), revealing significant gaps in current LLM capabilities for expert tasks.

**Strengths:**

1. The paper addresses a critical and widely acknowledged gap in LLM evaluation. Most benchmarks test for factual recall via multiple-choice or short-form answers. This work makes a significant contribution by shifting the evaluation to what experts actually do.

2. The CLEAR framework is a major strength. The idea of not comparing two long texts directly, but instead using an expert-designed rubric to decompose the task into a structured checklist, is a very strong and novel approach to "grounded" evaluation.

3. The paper's key findings are highly valuable. The primary result that even SOTA LLMs fail badly is a crucial data point for the community. More importantly, the discovery of a negative correlation between checklist coverage and F1 score is a critical insight, providing concrete evidence that models are adept at generating plausible, structured, but factually incorrect content, which is a key risk.

**Weaknesses:**

1. A weakness is the small number of samples. The benchmark contains 1050 samples total, about 100 samples/each. It is questionable whether 100 samples are sufficient to draw robust conclusions about a model's performance across an entire complex domain like law or medicine. This small scale could lead to noisy results.

2. The paper states that a portion of the benchmark is public. Given the small sample size (50-100 per task) and the highly unique, long-form nature of the prompts and data, these tasks are at a very high risk of being included in the training data of future models, which would quickly render the public benchmark obsolete for evaluation.

3. CLEAR relies on SOTA LLMs to function as "Checklist Mappers" and "Judges." This introduces a circular dependency where the evaluation of LLMs is dependent on the capabilities of another LLM. While Section 6 attempts to validate this, it's still a source of potential variance.

4. The benchmark's greatest strength (expert-designed rubrics) is also its greatest weakness. The paper notes that creating the rubric for a single task (T1) took over 10 hours of expert time. This creates a massive bottleneck, making the framework difficult and expensive to scale to new domains or even new tasks within the same domain.

**Questions:**

1. The main concern is the low sample size (100 per task). How did the authors validate that this sample size is statistically sufficient to be representative of the entire domain and to produce stable, reliable model rankings?

2. The CLEAR framework's results are contingent on the LLM used for mapping and judging (e.g., Qwen2.5-72B and GPT-4O). How sensitive are the final F1 scores (Table 2) to this choice? If you were to swap the judge/mapper to a different models (e.g., a Claude or Llama model), do the model rankings remain consistent?

3. Given the high cost of expert rubric design and the high risk of data contamination, what is the long-term vision for maintaining and expanding this benchmark? How can this approach scale without requiring hundreds of hours of domain-expert time for each new task?

4. The finding of high coverage but low F1 is fascinating. Do the authors have a hypothesis why this is happening? Is it a failure of long-context retrieval (the model fails to find the right facts in the 200K token input) or a failure of generation (the model finds the facts but hallucinates or misrepresents them to fit the required checklist structure)?

---

> ### Author Response · Authors · 2025-11-22
> **Response to Reviewer 15Gk (1/3)**
>
> Dear Reviewer 15Gk,
>
> We genuinely thank you for the time and effort you devoted to reviewing our paper. Your detailed and insightful suggestions are of great value to us as we work to improve the quality of our paper. In the following section, we address the concerns you raised:
>
> > **Question 1**: The main concern is the low sample size (100 per task). How did the authors validate that this sample size is statistically sufficient to be representative of the entire domain and to produce stable, reliable model rankings?
>
> **Response 1**:  To assess whether our sample size is sufficient to distinguish model performance, we conducted paired t-tests for each task. For every task, we compared the best-performing model against the second- and third-best models. The null hypothesis assumed no difference in performance between the model pair, while the alternative hypothesis tested whether the top model outperformed the comparison model.
>
> When comparing the best and second-best models, 5 out of 11 tasks showed statistically significant differences (p < 0.01), allowing us to reject the null hypothesis. Comparing the best and third-best models yielded even stronger differentiation, with 8 out of 11 tasks showing statistically significant differences (p < 0.01). These results suggest that our evaluation has enough statistical power to reliably distinguish meaningful performance gaps across models, providing confidence in the robustness of our reported rankings.
>
> > **Question 2**: The CLEAR framework's results are contingent on the LLM used for mapping and judging (e.g., Qwen2.5-72B and GPT-4O). How sensitive are the final F1 scores (Table 2) to this choice? If you were to swap the judge/mapper to a different models (e.g., a Claude or Llama model), do the model rankings remain consistent?
>
> **Response 2**: We thank the reviewer for this helpful suggestion and fully agree on the importance of assessing whether different evaluators produce consistent model rankings. To conduct this analysis, we are evaluating 8 models (Gemini-2.5-Pro, Gemini-2.0-Flash, GPT-4o, Mistral-Large-Instruct, Llama-3.1-8B-Instruct, Qwen2.5-7B, Qwen2.5-72B and Claude-3.5-Haiku) across five tasks (T1, T3, T4, T6, T8), using 3 evaluator models (Gemini-2.0-Flash, GPT-4o, and Qwen2.5-72B). These 8 models span a wide performance range, and the selected tasks cover a broad set of domains with varied input and output lengths.
>
> For each task, we compute performance using all 3 evaluators, derive evaluator-specific rankings, and then measure the Spearman rank correlation between GPT-4o’s rankings and those produced by the other evaluators to assess consistency. The table below reports these correlations for each task. The consistently high correlations indicate strong agreement across evaluators. This is expected because the evaluation focuses on measuring semantic overlap between the model-produced checklists and the ground-truth checklists which is a substantially easier task than solving the underlying expert tasks themselves. As a result, any sufficiently capable evaluator should yield highly consistent rankings.
>
> | Task | GPT-4o vs Gemini-2.0-Flash | GPT-4o vs Qwen2.5-72B | Gemini-2.0-Flash vs Qwen2.5-72B |
> |------|:-----------------------------:|:-------------------------:|:----------------------------------:|
> | T1   | 0.6429                      | 0.9762                  | 0.5952                           |
> | T3   | 0.9762                      | 0.8333                  | 0.7381                           |
> | T4   | 0.9524                      | 0.9762                  | 0.9762                           |
> | T6   | 0.8571                      | 1.0000                  | 0.8571                           |
> | T8   | 0.9524                      | 0.9286                  | 0.9524                           |

---

> > ### Author Response · Authors · 2025-11-22
> > **Response to Reviewer 15Gk (2/3)**
> >
> > > **Question 3**: Given the high cost of expert rubric design and the high risk of data contamination, what is the long-term vision for maintaining and expanding this benchmark? How can this approach scale without requiring hundreds of hours of domain-expert time for each new task?
> >
> > **Response 3**: We appreciate the reviewer’s question regarding the long-term sustainability of the benchmark, especially given the effort required for expert rubric construction and the importance of preventing data contamination.
> >
> > **Preventing contamination and ensuring long-term utility.** As noted in the footnotes on page 2, we will make it explicit on the benchmark website that all benchmark data must not be used for model training. Following standard practices of prior benchmarks, we structure ExpertLongBench into a public subset and a private subset, ensuring that evaluation remains reliable even as models evolve.
> >
> > Additionally, because we have collected more samples than as indicated. So it would allow us to periodically rotate and refresh the benchmark. If contamination or saturation is detected, affected samples will be replaced with new and more challenging instances as mentioned in the footnote on page 3. This establishes a dynamic mechanism that preserves the benchmark’s relevance and trustworthiness over time.
> >
> > **Reducing expert workload in rubric creation.** We acknowledge that constructing expert-level rubrics is non-trivial. Our long-term vision incorporates several strategies to reduce expert time while maintaining reliability:
> >
> > - **Protocol-refinement strategy** (mentioned in page 4 “Expert-guided Rubric Design” paragraph). Many practical domain-specific tasks are governed by well-established professional standards. For such tasks, rubric creation primarily involves adapting existing protocols rather than designing criteria from scratch, substantially reducing expert workload.
> > - **Human–LLM collaborative rubric design.** LLMs can assist experts by generating initial checklists or draft rubrics based on task descriptions and high-quality human reference outputs. Experts would then validate and refine these drafts, reducing the required expert hours while preserving domain rigor. Existing rubrics in our benchmark would further serve as in-context examples, enabling LLMs to propose increasingly accurate initial drafts over time.
> > - **Active collaboration with domain communities.** We plan to expand ExpertLongBench through multi-disciplinary, open collaboration, inviting domain researchers and practitioners to contribute tasks, provide feedback, and help refine rubrics. This shared stewardship further distributes the effort and enhances the benchmark’s coverage and credibility.
> >
> > We agree that expert benchmarking is inherently difficult and this is one of the reasons why our work is valuable. ExpertLongBench addresses a gap that simpler QA or multiple-choice datasets cannot: evaluating end-to-end, expert-level long-form generation. By combining careful task selection, protocol-based rubric design, human–LLM collaboration, and a dynamic refresh mechanism, we believe this benchmark can scale sustainably without requiring hundreds of expert hours for each new task.
> >
> > We appreciate the reviewer’s perspective and will incorporate these clarifications into the revised version.

---

> ### Author Response · Authors · 2025-11-22
> **Response to Reviewer 15Gk (3/3)**
>
> > **Question 4**: The finding of high coverage but low F1 is fascinating. Do the authors have a hypothesis why this is happening? Is it a failure of long-context retrieval (the model fails to find the right facts in the 200K token input) or a failure of generation (the model finds the facts but hallucinates or misrepresents them to fit the required checklist structure)?
>
> **Response 4**: We thank the reviewer for this insightful question. We agree that the pattern of high coverage but low F1 is intriguing and a warning sign for careful use of LLMs at expert domains. We view it as arising from a combination of retrieval challenges, long-context understanding challenges and lack of knowledge and reasoning capability in the target domain, depending on the task type.
>
> When the input is not extremely long, the bottleneck is the lack of knowledge and reasoning capability in the target domain. For tasks such as T7-ChemMDG (Molecule Description Generation) and T8-BioPDG (Protein Description Generation), the input length is moderate. In these settings, the model’s domain-specific reasoning and generation abilities remain weak. The model often hallucinates mechanistic details even though providing some content seems reasonable. These behaviors explain why coverage is high (the model identifies many relevant elements) but F1 remains low (the details are incorrect or improperly grounded).
>
> When the input is extremely long, retrieval and long-context understanding becomes an additional failure mode. For tasks with very long inputs (e.g., T2-LegalSFG, Statement of Facts Generation), retrieval and long-context understanding difficulties become more pronounced. As shown in Appendix I.1, we experimented with giving models a more detailed, rubric-augmented prompt. While this improved results modestly, performance remained very poor. We also show the analysis of the model's performance across different input and output lengths in Appendix D.3. Based on the results on Figure 8, the model tends to have better performance for low input length but the performance is still moderate. This shows models struggle to retrieve, understand, reason and synthesize reasonable output.
>
> Taken together, these results indicate that retrieval, long-context understanding and lack of knowledge and reasoning capability in the target domain contribute. We appreciate the reviewer’s thoughtful question and will incorporate this analysis into the revision.
>
> We hope this addresses your concern, and we would greatly appreciate it if you could consider updating your review score. Once again, thank you very much for your insightful and valuable suggestions!

---

> ### Author Response · Authors · 2025-11-28
>
> Dear Reviewer 15Gk,
>
> Thank you once again for the time and effort you devoted to reviewing our paper. As the rebuttal deadline is approaching, we would like to kindly confirm whether we have sufficiently addressed your concerns. Should there be any remaining questions or areas requiring further clarification, please do not hesitate to let us know. If you are satisfied with our responses, we would greatly appreciate your consideration in adjusting the evaluation scores accordingly.
>
> We sincerely look forward to your feedback.

---

### Official Review · Reviewer_kZKG · 2025-10-31

**Soundness:** 3
**Presentation:** 3
**Contribution:** 3
**Rating:** 6
**Confidence:** 3

**Summary:**

This paper introduces ExpertLongBench, a benchmark for expert-level generation with long input and long output, and a checklist-based evaluation method called CLEAR. The benchmark covers 11 domains, including law, science, medicine, education, and finance. Each task comes with an expert-written rubric and a reference that is mapped into a checklist, so an LLM judge can test whether a system answer covers each item. The authors provide a public subset, a private subset for held-out testing, code, and an evaluation pipeline. The goal is to measure factual coverage and faithfulness in realistic expert workflows rather than surface fluency.

Experimentally, the paper evaluates a broad set of LLMs on the suite and finds that even leading models perform far from perfect, with average F1 around 30. The tasks include very large inputs and long outputs, and the analysis shows a common pattern of high coverage but lower correctness. Human spot checks confirm the quality of the reference mapping on selected tasks. Judge reliability is studied by comparing GPT-4o with other judges, showing strong agreement and good correlation with an open model judge, and the pipeline supports using open models for the mapping step with similar rankings.

**Strengths:**

1. The authors present a benchmark with long inputs and outputs that requires expert-level knowledge, covering 11 domains. This is a valuable resource for the community.

2. The authors conduct comprehensive experiments on ExpertLongBench, including many frontier models, and show that the benchmark remains highly challenging even for top-performing models.

3. The authors devote substantial effort to data quality and model analysis, for example examining potential data contamination and designing an automated, relatively reliable evaluation pipeline.

**Weaknesses:**

1. **Checklist and rubric reliability.** My main concern is the quality of the checklists and rubrics, which the overall evaluation heavily relies on. While I acknowledge the authors’ efforts, many tasks appear to be curated by a single PhD student in a related field, which may be insufficient and could introduce bias. An ideal setup would involve multiple experts for curation, additional experts for independent review, and iterative updates until the experts reach a reliable agreement. I only noticed that T8 mentions experts “reached consensus,” without any quantitative agreement metrics. Introducing a more thorough process and reporting would improve confidence in the benchmark.

2. **“Long-form” positioning.** Although the paper claims a long-form generation benchmark, Table 1 shows that 8 of 11 tasks have inputs under 2k tokens and 9 of 11 tasks have references under 500 tokens. I understand this is already longer than many expert-level datasets, but compared with long-context-focused benchmarks (e.g., LongBench and L-Eval), it is less convincing to label the suite as a long-context benchmark.

3. **Missing references.** I recommend adding a comparison and citation to HealthBench. If the authors intend to position this work as a long-context benchmark, it would also help to compare with and cite LongBench and L-Eval, which include long-form generation tasks.

**Questions:**

See the above.

---

> ### Author Response · Authors · 2025-11-22
> **Response to Reviewer kZKG (1/2)**
>
> Dear Reviewer kZKG,
>
> We genuinely thank you for the time and effort you devoted to reviewing our paper. Your suggestions are of great value to us as we work to improve the quality of our paper. In the following section, we address the concerns you raised:
>
> > **Weakness 1**: Checklist and rubric reliability. My main concern is the quality of the checklists and rubrics, which the overall evaluation heavily relies on. While I acknowledge the authors’ efforts, many tasks appear to be curated by a single PhD student in a related field, which may be insufficient and could introduce bias. An ideal setup would involve multiple experts for curation, additional experts for independent review, and iterative updates until the experts reach a reliable agreement. I only noticed that T8 mentions experts “reached consensus,” without any quantitative agreement metrics. Introducing a more thorough process and reporting would improve confidence in the benchmark.
>
> **Response 1**: We thank the reviewer for raising this important point regarding the reliability of our checklists and rubrics. We agree that expert involvement and clear reporting are essential for ensuring high-quality evaluation. Below, we clarify how our tasks and rubrics were designed to minimize ambiguity and why our current setup offers reliable, domain-grounded assessment.
>
> **Well-defined tasks with low ambiguity.** Most tasks in our benchmark are well-established domain-specific tasks where the criteria for what constitutes a “good” output are well defined and widely agreed upon within each field. These tasks involve minimal, if any,  subjective or open-ended judgments; rather, they follow standard domain conventions. As a result, the space of reasonable evaluation criteria is narrow, and the potential for annotator bias is limited.
>
> **Rubrics grounded in authoritative references.** For the majority of tasks (T1, T4–T11), the rubrics were constructed based on **existing reference resources** or authoritative domain standards. Appendix B provides the details of evaluation rubrics construction for every task. These rubrics capture the core aspects that practitioners consistently use when assessing and producing outputs, reducing variance in what evaluators consider important. For other tasks: T2 (Legal Task): The rubric was created by two JD students and reviewed and confirmed by one legal professor familiar with this specific task. T3 (Material Science Task): The rubric was confirmed by a senior PhD student specializing in solid-state synthesis, with extensive research experience and multiple publications in the area. The task itself represents a fundamental material science procedure with well-established assessment criteria.
>
> Because these tasks are highly structured and the reference solutions are themselves reliable resources, a domain-trained PhD student is sufficiently qualified to identify whether a model output satisfies the essential criteria.
>
> **Verification of rubric reliability using human references.** We further validated rubric reliability by analyzing human references output of the task. This analysis demonstrates that the rubrics accurately capture key quality dimensions.
>
> **On consensus and agreement metrics.** For T8, we stated that domain experts “reached consensus”; we agree that this could benefit from more explicit reporting, and we will revise the wording to clarify the process. Specifically, the two PhD students who designed the task rubric independently reviewed relevant domain literature and designed the rubric items, then discussed any discrepancies in their initial checklist interpretations. The discrepancies arise mainly from differing interpretations of which checklist items are mandatory versus conditionally applicable. Consensus was reached through iterative discussion grounded in published scientific criteria. We will update the paper to reflect this process clearly. However, given the deterministic nature of these tasks and the low ambiguity of the required outputs, the variance across checklist items is inherently limited.
>
> Finally, we would like to restate that our benchmark aims to move **beyond traditional QA and multiple-choice evaluations** toward end-to-end expert-level tasks. Designing rubrics for such tasks is inherently more challenging, and our work represents an important step toward establishing systematic, reproducible evaluation in these real-world domains.
>
> We appreciate the reviewer’s suggestions and will incorporate more explicit reporting of rubric verification in the next revision.

---

> > ### Author Response · Authors · 2025-11-22
> > **Response to Reviewer kZKG (2/2)**
> >
> > > **Weakness 2**: “Long-form” positioning. Although the paper claims a long-form generation benchmark, Table 1 shows that 8 of 11 tasks have inputs under 2k tokens and 9 of 11 tasks have references under 500 tokens. I understand this is already longer than many expert-level datasets, but compared with long-context-focused benchmarks (e.g., LongBench and L-Eval), it is less convincing to label the suite as a long-context benchmark.
> >
> > **Response 2**: We thank the reviewer for the thoughtful clarification. We'll clarify the focus is on expert level tasks that require long output, and they often come with longer context as well. But this comes as natural distribution, unlike existing long-context benchmarks, the long contexts are often synthetically constructed. We appreciate the reviewer’s observation and will clarify this in the paper.
> >
> > > **Weakness 3**: Missing references. I recommend adding a comparison and citation to HealthBench. If the authors intend to position this work as a long-context benchmark, it would also help to compare with and cite LongBench and L-Eval, which include long-form generation tasks.
> >
> > **Response 3**: Thank you for your suggestions! As mentioned in the previous comment, we'll mention these suggested works and clarify our focus is not on long-context generation but long-form generation, where outputs are open-ended and consist of content richer than popular expert-level benchmarks.  And we'll discuss healthbench, note that they construct rubric at instance-level but it does not contain a reference for each instance.
> >
> > We hope this addresses your concern, and we would greatly appreciate it if you could consider updating your review score. Once again, thank you very much for your valuable comments and suggestions!

---

> ### Author Response · Authors · 2025-11-28
>
> Dear Reviewer kZKG,
>
> Thank you once again for the time and effort you devoted to reviewing our paper. As the rebuttal deadline is approaching, we would like to kindly confirm whether we have sufficiently addressed your concerns. Should there be any remaining questions or areas requiring further clarification, please do not hesitate to let us know. If you are satisfied with our responses, we would greatly appreciate your consideration in adjusting the evaluation scores accordingly.
>
> We sincerely look forward to your feedback.

---

### Official Review · Reviewer_ejKy · 2025-11-04

**Soundness:** 2
**Presentation:** 3
**Contribution:** 3
**Rating:** 4
**Confidence:** 4

**Summary:**

This paper introduces EXPERTLONGBENCH, an expert-level benchmark with 11 tasks across 9 domains, focusing on realistic long-form expert workflows (e.g., legal case summarization, clinical note generation) that require outputs exceeding 5,000 tokens. Each task includes an expert-designed/validated rubric and human-written references for grounded evaluation.
It also proposes CLEAR, an evaluation framework that extracts checklists from model outputs and references based on task rubrics, then compares checklist items for fine-grained, objective assessment. Open-weight models like Qwen2.5-72B work effectively in CLEAR, enabling low-cost, reproducible evaluation.
Benchmarking 13 LLMs shows top model Gemini-2.5-Pro only achieves an average F1 of 33.4, revealing significant gaps. Critically, high checklist item coverage (irrespective of correctness) correlates negatively with F1—models often produce content seemingly aligned with domain standards but incorrect, posing misleading risks. This work fills gaps in expert-level long-form LLM evaluation and guides future model improvement.

**Strengths:**

1. This work fills the gap in expert-level long-form generation evaluation, which appears to be of considerable significance.
2. This work proposes CLEAR, a method that extracts checklists from reference outputs for comparison. This approach exhibits higher accuracy compared to existing rubric-based methods, and the experiments also demonstrate that models may potentially game such rubric-based approaches.

**Weaknesses:**

1. The long-form evaluation method is excessively costly. As shown in Table 1, many tasks involve input exceeding 100,000 tokens. For a single task with 100 samples, 10 million tokens are required—and this only accounts for input, excluding the cost of completion. The authors put their cost in Section G.3, where the evaluation costs them over $1000.  The practicality of evaluation at this scale is questionable.
2. In terms of evaluation, while it assesses long-form generation capabilities, it omits consideration of various solutions for long-form text processing, such as RAG (retrieval-augmented generation) approaches or the integration of agentic workflows (e.g., preliminary summarization). In real-world scenarios, it is uncommon to directly feed such long prompts to models for generation. This aspect lacks practical relevance.
3. While the paper validates GPT-4o’s consistency with another closed model (Gemini-2.0-Flash, Cohen’s Kappa: 0.81–0.89, Section 4.2) and verifies checklist extraction quality via human inspection (T1/T6: 95.12–99.99% faithfulness, Section E.1), it does not compare GPT-4o’s evaluation results with human expert judgments. The authors acknowledge this gap in Section I.3 (Limitations), noting that “LLM-based evaluation can yield erroneous assessments and requires careful human oversight”—which echoes your concern about unvalidated reliance on GPT-4o.
4. A portion of the data is not publicly available, which hinders the community from conducting tests and follow-up research.

**Questions:**

NA

---

> ### Author Response · Authors · 2025-11-22
> **Response to Reviewer ejKy (1/2)**
>
> Dear Reviewer ejKy,
>
> We genuinely thank you for the time and effort you devoted to reviewing our paper. In the following section, we address the concerns you raised:
>
> > **Weakness 1**: The long-form evaluation method is excessively costly. As shown in Table 1, many tasks involve input exceeding 100,000 tokens. For a single task with 100 samples, 10 million tokens are required—and this only accounts for input, excluding the cost of completion. The authors put their cost in Section G.3, where the evaluation costs them over $1000. The practicality of evaluation at this scale is questionable.
>
> **Response 1**: Thank you for the thoughtful concern regarding evaluation cost. We respectfully clarify that our actual evaluation expenses are substantially lower than the reviewer’s estimate. As reported in Table 56, the total cost for generating model outputs on our benchmark is only **$ 36.97** for GPT-5 and **$ 31.38** for Gemini-2.5-Pro. For the **evaluation itself**, Table 57 shows that the full checklist-based evaluation across **13 models** costs **$631.55**, averaging **$48.58 per model**. If considering only the **public portion** of the benchmark, the average cost further decreases to **$26.72 per model**, which demonstrates that long-form evaluation is not prohibitively expensive in practice.
>
> Moreover, Section E.3 demonstrates that **open models can serve as effective judges**, offering an even more cost-efficient alternative. In particular, Qwen2.5-72B achieves a correlation of 0.876 with GPT-4o, indicating that open-model judges can provide highly reliable evaluations at **zero proprietary-model cost**. This suggests that, even under tight budgets, scalable long-form evaluation remains feasible and practical.
>
>
> > **Weakness 2**: In terms of evaluation, while it assesses long-form generation capabilities, it omits consideration of various solutions for long-form text processing, such as RAG (retrieval-augmented generation) approaches or the integration of agentic workflows (e.g., preliminary summarization). In real-world scenarios, it is uncommon to directly feed such long prompts to models for generation. This aspect lacks practical relevance.
>
> **Response 2**:  Thank you for this thoughtful suggestion. Our original goal was to evaluate whether natively deployed LLMs possess sufficient expert knowledge to independently solve expert-level practical tasks. In response to your feedback, we implemented a RAG-based agent using GPT-5 with langchain that sequentially retrieves and integrates relevant information from the provided document context to address each task. We implemented retrieval by embedding all document paragraphs using OpenAI’s text-embedding-3-large model. At each iteration, GPT-5 generates a query, which is then used to retrieve the most relevant paragraphs; these retrieved passages are subsequently fed back into GPT-5 to support reasoning and task completion.  Because Tasks T1 and T2 involve particularly long documents, where retrieval-augmented methods are especially applicable, we applied this RAG approach to these tasks. The evaluation results (F1-scores) are presented below. We find that the RAG agent underperforms, suggesting that full access to the global document context is crucial for these tasks. Although this generic RAG implementation did not yield performance gains, we believe that a domain- and task-specific retrieval and reasoning strategy could lead to stronger results. However, developing such specialized systems is beyond the scope of our current work.
>
> | Task | GPT-5  | GPT-5 RAG Agent |
> |:----:|:-----:|:----------------:|
> | T1   | 27.15 | 24.27           |
> | T2   | 10.34 | 4.85           |

---

> > ### Author Response · Authors · 2025-11-22
> > **Response to Reviewer ejKy (2/2)**
> >
> > > **Weakness 3**: While the paper validates GPT-4o’s consistency with another closed model (Gemini-2.0-Flash, Cohen’s Kappa: 0.81–0.89, Section 4.2) and verifies checklist extraction quality via human inspection (T1/T6: 95.12–99.99% faithfulness, Section E.1), it does not compare GPT-4o’s evaluation results with human expert judgments. The authors acknowledge this gap in Section I.3 (Limitations), noting that “LLM-based evaluation can yield erroneous assessments and requires careful human oversight”—which echoes your concern about unvalidated reliance on GPT-4o.
> >
> > **Response 3**: We thank the reviewer for raising this important point. We agree that validating LLM-as-a-judge against human expert judgments is critical for establishing trust in the evaluation pipeline. Following the reviewer’s suggestion, we conducted an additional human evaluation to directly compare GPT-4o’s rubric-based assessments with domain-expert ratings.
> >
> > We randomly sampled 250 evaluation instances from two representative domain-specific tasks: T7 (ChemMDG: Molecule Description Generation) and T8 (BioPDG: Protein Description Generation), with 100 instances for T7 and 150 instances for T8. For each task, we randomly selected outputs from two models (GPT-5 and Gemini-2.5-Pro). Two PhD students and one graduate student who originally contribute to the design of the rubrics for these tasks independently evaluate the sampled outputs. To ensure a fair and controlled comparison, the human evaluators follow the similar evaluation prompt GPT-4o used as evaluation guideline.
> >
> > Evaluation results show that the accuracy, which measures the percentage of outputs where GPT-4o’s binary rubric-based judgments agree with the human experts, is 91.3% for T8 and 92% for T7. We appreciate the reviewer's suggestion and will add the corresponding evaluation information and results to the revised paper.
> >
> > > **Weakness 4**: A portion of the data is not publicly available, which hinders the community from conducting tests and follow-up research.
> >
> > **Response 4**: Thank you for your comment. To address concerns about data openness and contamination, we have intentionally designed the benchmark with both a public and a private subset. The public portion is fully accessible and sufficient for initial experimentation and development. The private subset is deliberately held back due to the critical nature of the domain, the sensitivity of the data involved, and to reduce the risk of data contamination. This separation allows us to better evaluate models’ generalization capabilities  and avoid overfitting to publicly available datasets. To ensure continued accessibility, we provide a Model Submission Form through which researchers can evaluate their models on the private subset. This process is described on our benchmark website and noted in the footnote 3 on Page 4 of the paper. This design allows us to better assess generalization and prevent models from inadvertently optimizing on test data seen during pretraining.
> >
> > We hope this addresses your concern, and we would greatly appreciate it if you could consider updating your review score. Once again, thank you very much for your valuable comments and suggestions!

---

> ### Author Response · Authors · 2025-11-28
>
> Dear Reviewer ejKy,
>
> Thank you once again for the time and effort you devoted to reviewing our paper. As the rebuttal deadline is approaching, we would like to kindly confirm whether we have sufficiently addressed your concerns. Should there be any remaining questions or areas requiring further clarification, please do not hesitate to let us know. If you are satisfied with our responses, we would greatly appreciate your consideration in adjusting the evaluation scores accordingly.
>
> We sincerely look forward to your feedback.

---

### Meta-Review · Area_Chair_BHmj · 2025-12-28

**Summary:**

This paper introduces ExpertLongBench, a benchmark of  long-form generation tasks (11 tasks / 9 domains) with expert-designed rubrics, and proposes CLEAR, a rubric-to-checklist evaluation pipeline. Empirically, the benchmark appears meaningfully challenging for current frontier models: the best reported model (Gemini-2.5-Pro) achieves only ~33.4 average F1, with some tasks extremely low across all systems, suggesting the dataset is not saturated and provides a useful stress test beyond standard short-form QA leaderboards.

The key reviewer concerns centered on (i) practicality/cost and whether agentic/RAG settings should be included, (ii) reliability and scalability of expert rubrics/checklists, (iii) evaluator circularity and checklist-mapping validity. Overall, the rebuttal added meaningful experiments/analyses that address the largest practicality and validation concerns, and remaining issues are mostly about practices of validation/reporting rather than fundamental flaws.

**Reviewer Concerns:**

Concerns addressed by the rebuttal / revisions (largely resolved)

Need for RAG / agentic baseline: authors added an agentic/RAG-style experiment on the long-document tasks and reported results, which addresses “missing baseline” and helps interpret whether long-context ingestion is actually necessary for these tasks.
Human validation of LLM-as-a-judge: authors added a human expert comparison study on two tasks, reporting high agreement with the rubric-based LLM judgments.
Evaluator dependence: authors provided cross-evaluator agreement/ranking stability analyses.

Concerns partially addressed, still outstanding

Checklist mapping is still the main risk. CLEAR relies on accurate extraction/mapping of rubric items into checklist evidence before judging. While the paper provides mapping validation on subsets (e.g., faithfulness checks for certain tasks), the most convincing quantitative mapping validation is not shown uniformly across all 11 tasks.

**Reviewer Scores:**

Most reviewers are leaning positive about the work. LZpR/kZKG/15Gk might maintain or increase due to many added results and validation of evaluation pielines.

---

### Decision · Program_Chairs · 2026-01-26

Accept (Poster)